# Assessment of the Impact of Coffee Waste as an Alternative Feed Supplementation on Rumen Fermentation and Methane Emissions in an In Vitro Study

**Belgutei Batbekh [1], Eslam Ahmed [2,3], Masaaki Hanada [2], Naoki Fukuma [2,4] and Takehiro Nishida [2,\*]**

[1] Graduate School of Animal Husbandry, Obihiro University of Agriculture and Veterinary Medicine, Inada, Obihiro 080-8555, Japan; belguteibatbekh@gmail.com
[2] Department of Life and Food Sciences, Obihiro University of Agriculture and Veterinary Medicine, Inada, Obihiro 080-8555, Japan; eslam_kh@obihiro.ac.jp (E.A.); hanada@obihiro.ac.jp (M.H.); n.fukumax@obihiro.ac.jp (N.F.)
[3] Department of Animal Behavior and Management, Faculty of Veterinary Medicine, South Valley University, Qena 83523, Egypt
[4] Research Center for Global Agromedicine, Obihiro University of Agriculture and Veterinary Medicine, Inada, Obihiro 080-8555, Japan
[\*] Correspondence: nishtake@obihiro.ac.jp

**Abstract:** Spent coffee waste is the most common by-product of coffee processing, and it has the potential to be used as a source of organic compounds for ruminant diets. The objective of this study was to evaluate the optimal inclusion level and method for using spent coffee waste (SCW) as a ruminant feed and investigate its effects on rumen fermentation characteristics and methane ($CH_4$) production. The present in vitro batch culture study was conducted using two different experimental designs. The first experimental design (TRIAL. 1) was performed using a control diet of 500 mg of fresh matter basal diet (60% hay/40% concentrate), with SCW being used as a feed additive at 1%, 10% and 20% of the substrate. The second experimental design was performed using the same control diet, with spent coffee waste replacing either part of the hay (TRIAL. 2) or some of the concentrate mixture (TRIAL. 3) at four different dosages (30:70, 50:50, 70:30 and 100). When SCW was supplemented as a feed additive, there were increases in the production of volatile fatty acids and gas; however, it did not show any suppressive effects on $CH_4$ production. In contrast, when SCW was included as a replacement for hay or concentrate, there were significant reductions in $CH_4$ production with increasing levels of SCW inclusion. These reductions in $CH_4$ production were accompanied by negative effects on nutrient digestibility and total volatile fatty acid production. These findings demonstrate that SCW could potentially be used as a prebiotic feed additive. Additionally, when SCW is used as a replacement for silage at 70:30 and 50:50 dosages appear to be feasible as a substitute for animal feed (hay and concentrate).

**Keywords:** by-products; methane emission; rumen fermentation; spent coffee waste; sustainability



## 1. Introduction

The world population is expected to reach approximately 9.7 billion people by 2050 and 10.4 billion people by 2100 [1]. This growing population needs to be fed, so the prevalence of animal products, such as meat and milk, in human diets needs to be considered [2]. Ruminants play an important role in animal production and contribute significantly to the overall quantity of animal products on the market. In the future, there will be an increased demand for ruminants in order to meet the food requirements of the growing global population. However, ruminants are also the most significant contributors to greenhouse gas (GHG) emissions, particularly methane ($CH_4$). When their numbers increase, this has a significant effect on global warming [3]. In addition, the production of $CH_4$ during the fermentation process in the rumen is correlated with the loss of energy from the

consumed feed. Moreover, $CH_4$ is 25 times more potent than $CO_2$ in terms of trapping heat from the sun [4]. Therefore, reducing $CH_4$ emissions from ruminants would significantly decrease the associated environmental impacts, as long as energy utilization efficiency is not affected [5]. Thus, there is a need to find strategies that improve feed efficiency, balance the supply of nutrients to meet animal requirements, reduce environmental impacts and achieve economic benefits [3]. So, it is important for animal nutrition researchers to focus on finding alternative options to replace conventional resources and feed additives [6].

Accordingly, using by-products from human food as feed for livestock is considered to be a possible solution. Coffee is one of the most popular beverages in the world, and several by-products are generated throughout its processing stages. One of the major by-products of coffee is spent coffee waste (SCW), which contains large amounts of organic compounds, particularly lipids, polyphenols and polysaccharides [7]. Due to the presence of these bioactive compounds, SCW is used in several industries, including biodiesel, cosmetics, construction and animal feed [8–12]. Moreover, the utilization of spent coffee waste as an alternative feed source for ruminants could help to mitigate $CH_4$ emissions, reduce waste and improve the environmental sustainability of livestock production; however, the dosages, processing methods and effects on animal health and performance need to be carefully considered.

Previously, researchers [7,11,13,14] have extensively explored the potential of raw or ensiled spent coffee waste (SCW) as a feed source for ruminants. Their investigations revealed that SCW offers both advantages and disadvantages, with the outcomes depending significantly on the dosage. Specifically, the presence of compounds like polyphenols, which are abundant in SCW, has been shown to have a substantial impact. At higher dosages, polyphenols can hinder nutrient digestibility, potentially affecting animal health and production. Moreover, elevated levels of fatty acids in SCW, particularly at higher concentrations, have their own set of challenges. These fatty acids can alter the feed's overall nutritional composition, potentially impacting ruminant performance and health. Therefore, a nuanced understanding of the dosage effects of polyphenols and fatty acids is crucial when considering the inclusion of SCW in ruminant diets. Additionally, the use of SCW as a feed source for ruminants depends on a number of factors, such as cost, availability, processing and compatibility with other dietary components. Furthermore, appropriate processing methods (such as ensiling) need to be determined to mitigate any negative effects of compounds on animal performance and nutrient utilization. Large quantities of high-moisture by-products are produced in many countries, including Japan; therefore, there is the need to develop technologies to design superior animal feed using SCW and enable the long-term storage of the resulting silage [15,16]. In Japan, there is an increasing interest in making silage by mixing wet and dry by-products, which offers a number of advantages, such as the reduced risk of effluent production, stabilized rumen function and extended storage periods [15]. The addition of lactic acid and soybean curd to silage when ensiling it with fresh grass or certain vegetable residues can improve fermentation quality. Moreover, when mixed with silage, these additives can also increase dry matter digestibility and reduce ruminal $CH_4$ production [16,17]. Therefore, the objective of this in vitro study was to assess the impact of using raw or ensiled SCW as a feed additive or a partial replacement for the basal components (hay or concentrate) in ruminant diets on rumen fermentation profiles and $CH_4$ production. Moreover, it is also important to establish the optimal level of SCW in animal diets [11,14]. However, there are still limitations to the potential use of SCW as a feed additive or replacement for conventional feed and the exact optimal dosages and methods remain unclear. Our hypothesis for Trail 1 was that the addition of SCW would improve rumen fermentation characteristics and mitigate $CH_4$ emissions. In Trails 2 and 3, we hypothesized that SCW would effectively replace the conventional feed ingredients in ruminant diet without adverse impact on rumen fermentation profile.

## 2. Materials and Methods

### 2.1. Basal Diet and Spent Coffee Waste

The basal diet consisted of ground Kleingrass (*Panicum coloratum*) hay with a particle size of 1 mm and a concentrate mixture (Alpha-Kotan, Chubu Shiryo Co., Ltd., Nagoya, Aichi, Japan). The SCW, both raw and ensiled, was provided in powder form by Sanyu Group Co., Ltd., Sagamihara City, Kanagawa, Japan. The chemical compositions of the SCW and basal diet components are detailed in Table 1.

**Table 1.** Chemical composition of the feed used in this study.

| (g/kg Dry Matter) | Klein Grass | Concentrate | Coffee (Raw) | Coffee (Silage) |
|---|---|---|---|---|
| Dry Matter (g/kg in fresh matter) | 908.6 | 884.4 | 961.3 | 956.6 |
| Organic Matter | 909.1 | 943.6 | 978.2 | 961.8 |
| Crude ash | 87.2 | 52.8 | 17.6 | 34.5 |
| Crude Protein | 140.8 | 180.0 | 126.0 | 154.9 |
| Ether Extract | 20.6 | 35.0 | 140.0 | 146.3 |
| NDF [1] | 696.2 | 524.4 | 708.0 | 792.9 |
| ADF [2] | 366.7 | 97.0 | 442.4 | 422.8 |
| ADL [3] | 89.6 | 20.0 | 232.2 | 234.0 |

[1] NDF = neutral detergent fiber. [2] ADF = acid detergent fiber. [3] ADL = acid detergent lignin.

### 2.2. Preparation of the Silage

The silage preparation was conducted at Sanyu Group Co., Ltd., Sagamihara City, Kanagawa, Japan. Spent coffee waste was obtained from Starbucks coffee stores across Japan. After being stored at the Customer Futures Distribution Center by Starbucks' chilled logistics, samples were collected at the factory by Sanyu Group logistics. After being drained at the stores, the spent coffee waste was packed in plastic bags, sprayed with vinegar spray and sealed for storage. The collected substrates were mixed with dried bean curd, bran, soy sauce dregs, vinegar and lactic acid bacteria. Then, the mixture was put in the polyethylene bags and placed into a stainless steel container for incubation. The entire ensiling process lasted for 14 days and was performed in the Sanyu Group factory.

### 2.3. Rumen Fluid Collection

The experimental animals for this study were housed and cared for at the Field Science Center, Obihiro University of Agriculture and Veterinary Medicine, Japan. The animal management and sampling procedures were approved by the Obihiro University of Agriculture and Veterinary Medicine's Animal Care and Use Committee (Approval number: 21-212).

In this study, two rumen-fistulated, non-lactating Holstein cows, which were about 9 years old, were used as rumen fluid donor animals. The cows were fed at maintenance level on a diet of orchard grass (*Dactylis glomerata*) hay (organic matter (OM), 980 g/kg; crude protein (CP), 132 g/kg; neutral detergent fiber (NDF), 701 g/kg; acid detergent fiber (ADF), 354 g/kg; acid detergent lignin (ADL), 40 g/kg; dry matter (DM) base), with free access to clean drinking water and mineral blocks (Koen® SELENICS TZ, Nippon Zenyaku Kogyo Co., Koriyama, Fukushima, Japan). About 1.3 L of rumen fluid was collected from 4 different places in rumen of both cows, and the then strained fluid was placed into a pre-warmed Thermos flask. The collected rumen fluid was immediately transferred to the laboratory within 15 min.

### 2.4. Experimental Design

This study was conducted using three experimental designs. The first experimental design (TRIAL. 1) was performed using a control diet (control group) of 500 mg of fresh matter basal diet (60% hay/40% concentrate). The SCW (both raw and ensiled) was added directly to the bottles (outside the nylon bag) and used as a feed additive at 1%, 10%, and 20% of the substrate. In this trial, using the raw and ensiled SCW were conducted separately. The second (TRIAL. 2) and the third (TRIAL. 3) experimental designs were con-ducted

using the same control diet as TRIAL. 1, but the SCW (both raw and ensiled) was included in the basal diet (in the nylon bag) to replace either hay or concentrate. TRI-AL. 2 and TRIAL. 3 were carried out on different days. TRIAL. 2 focused on replacing part of the hay with SCW, while TRIAL. 3 examined the replacement of a proportion of the concentrate with SCW. In TRIAL. 2, four different dosages of SCW (raw and ensiled) were included in the basal diet to replace the hay at different inclusion levels: 70:30 (42% hay/18% SCW/40% concentrate); 50:50 (30% hay/30% SCW/40% concentrate); 30:70 (18% hay/42% SCW/40% concentrate); and 100 (60% SCW/40% concentrate). In TRIAL. 3, another four dosages of SCW (raw and ensiled) were used to replace a proportion of the concentrate as follows: 70:30 (60% hay/28% concentrate/12% SCW); 50:50 (60% hay/20% concentrate/20% SCW); 30:70 (60% hay/12% concentrate/28% SCW); and 100 (60% hay/40% SCW). In TRIAL. 1, each group had four replicates and the experiment was repeated on four separate days. In TRIAL. 2 and TRIAL. 3, each group had three replicates and the experiments were repeated on three different days. In all of the trials, each run included two bottles for blank.

### 2.5. In Vitro Incubation Procedure

In the present study, 500 mg of the substrate was added to pre-weighed and nylon bags that has a fixed size and a pore size of $53 \pm 10$ μm (BG1020, Sanshin Industrial Co., Ltd., Yokohama, Kanagawa, Japan). These bags were sealed using a heat-sealer and then placed into 120 mL glass fermentation bottles. Via continuous $CO_2$ flushing, 40 mL of artificial saliva [18] and 20 mL of rumen fluid were added to each fermentation bottle. The bottles were then reinjected with $CO_2$ before being sealed with rubber and aluminum caps (Maruemu Co., Ltd., Osaka, Japan). The incubation procedure was as described by Ahmed et al. (2022) [19].

After 24 h of incubation, total gas production was measured using a gas-tight syringe, and headspace gas was collected from each bottle and stored in a vacuum tube (BD Vacutainer, Becton Drive, Franklin Lakes, NJ, USA). Then, the gas composition was analyzed via gas chromatography (GC-8A, Shimadzu Corp., Kyoto, Japan), as described previously by Ahmed et al. (2022) [19]. Next, the bottles were opened, the pH was measured immediately using a pH meter (LAQUA F-72, HORIBA Scientific, Kyoto, Japan), and 1 mL of the culture medium was collected in an Eppendorf tube (Eppendorf AG, Hamburg, Germany) and centrifuged at $16,000 \times g$ at 4 °C for 5 min. Following the centrifugation, the supernatant was gathered for further volatile fatty acid (VFA) analysis, which was measured via high-performance liquid chromatography (Shimadzu LC-20 HPLC, Shimadzu Corp., Kyoto, Japan). To determine the in vitro dry matter digestibility (IVDMD), the nylon bags containing the substrate were rinsed with tap water until the effluent became clear. They were then dried at 60 °C for 48 h to enable us to measure the IVDMD, which was calculated as the percentage of DM that disappeared from the initial DM weight that was input into the bags.

### 2.6. Chemical Analysis

The chemical composition analyses of the SCW, hay and concentrate mixture were performed according to the Association of Official Analytical Chemists procedures [20]. The DM content was determined by drying the matter in an oven at 135 °C for 2 h (930.15). The OM and ash contents were measured by placing the samples in a muffle furnace at 500 °C for 3 h (942.05). Nitrogen (N) content was measured according to the method of Kjeldahl (984.13) using an electrical heating digester (DK 20, VELP Scientifica, Usmate (MB), Monza, Italy) and an automatic distillation apparatus (UDK 129 VELP Scientifica, Usmate (MB), Monza, Italy), and CP was then estimated as $N \times 6.25$. The NDF and ADF contents were estimated and expressed as the inclusive residual ash values using an ANKOM200 fiber analyzer (Ankom Technology, Methods 6 and 5, respectively; ANKOM Technology Corp., Macedon, NY, USA). NDF content was measured using sodium sulfite without heat-stable α-amylase (FSS, ANKOM Technology).

### 2.7. Statistical Analysis

All data were analyzed using SAS version 9.4 (SAS Institute Inc., Cary, NC, USA). For all experiments, the data were analyzed using PROC MIXED models, including the treatments as fixed effects, whereas the experimental runs were considered random effects. The values are presented as the means with the pooled standard errors of the means. Any differences in means between the experimental groups were estimated using Tukey's test. Statistical significance difference was declared at $p < 0.05$, and a tendency was noted when $p$-value was between 0.05 and 0.10.

## 3. Results

### 3.1. TRIAL. 1

The inclusion of both the raw and ensiled SCW as an additive at 1%, 10% and 20% levels resulted in increased total gas production (8.7–11.8%) compared to the control diet (Tables 2 and 3). The IVDMD was significantly lower ($p < 0.008$) with the addition of raw SCW at the 10% and 20% levels compared to the control diet (3.4–12.06%; Table 4), while ensiled SCW did not show any significant effects on digestibility (Table 5). The total VFA production was significantly increased in the raw SCW groups ($p < 0.04$, Table 4). Additionally, the raw SCW groups showed increases in propionate production, while the ensiled SCW groups demonstrated increased butyrate production (Tables 4 and 5). Notably, none of groups showed any significant effects on pH or acetate to propionate (A:P) ratio (Tables 4 and 5). However, more significant effects on rumen parameters were observed when raw SCW was used as a feed additive.

**Table 2.** Effect of raw SCW as feed additive on gas production profile.

| Item | Control | 1% | 10% | 20% | SEM | *p*-Value |
|---|---|---|---|---|---|---|
| Total gas (mL/day) | 52.9 [c] | 56.0 [ab] | 57.5 [ab] | 58.7 [a] | 1.26 | <0.001 |
| Total gas/DDM [1] (mL/g) | 108.1 [b] | 111.5 [b] | 125.0 | 153.8 [a] | 6.08 | 0.001 |
| $CH_4$ (%) | 5.2 | 4.8 | 4.8 | 4.9 | 0.07 | 0.060 |
| $CO_2$ (%) | 94.8 | 95.2 | 95.2 | 95.1 | 0.07 | 0.060 |
| $CH_4$/DDM (mL/g) | 5.5 [bc] | 5.3 [bc] | 5.9 [b] | 7.5 [a] | 0.26 | 0.004 |
| $CH_4$ (mL/day) | 2.7 | 2.7 | 2.8 | 2.9 | 0.07 | 0.150 |
| $CO_2$/DDM (mL/g) | 102.5 [bc] | 106.2 [bc] | 119.0 [b] | 146.3 [a] | 5.83 | 0.001 |
| $CO_2$ (mL/day) | 50.2 [c] | 53.3 [ab] | 54.8 [ab] | 55.8 [a] | 1.21 | <0.001 |

[1] DDM, Digestible dry matter. SEM: Standard error of the mean. [a,b,c] means in the same row with different superscript differ significantly ($p < 0.05$).

**Table 3.** Effect of ensiled SCW as feed additive on gas production profile.

| Item | Control | 1% | 10% | 20% | SEM | *p*-Value |
|---|---|---|---|---|---|---|
| Total gas (mL/day) | 56.5 [bc] | 54.5 [c] | 61.7 [ab] | 63.2 [a] | 1.14 | 0.003 |
| Total gas/DDM [1] (mL/g) | 103.2 [ab] | 97.3 [b] | 112.1 [ab] | 118.9 [a] | 2.80 | 0.010 |
| $CH_4$ (%) | 4.0 [c] | 7.0 [a] | 5.7 [a–c] | 6.0 [ab] | 0.35 | 0.005 |
| $CO_2$ (%) | 96.0 [a] | 93.0 [bc] | 94.3 [ab] | 94.0 [bc] | 0.35 | 0.005 |
| $CH_4$/DDM (mL/g) | 4.1 [c] | 6.8 [ab] | 6.4 [a–c] | 7.2 [a] | 0.40 | 0.011 |
| $CH_4$ (mL/day) | 2.2 [b] | 3.8 [a] | 3.5 [ab] | 3.8 [a] | 0.21 | 0.009 |
| $CO_2$/DDM (mL/g) | 99.1 [ab] | 90.5 [b] | 105.7 [ab] | 111.1 [a] | 2.65 | 0.010 |
| $CO_2$ (mL/day) | 54.3 [a–c] | 50.7 [c] | 58.2 [ab] | 59.4 [a] | 1.06 | 0.002 |

[1] DDM, Digestible dry matter. SEM: Standard error of the mean. [a,b,c] means in the same row with different superscript differ significantly ($p < 0.05$).

**Table 4.** Effect of raw SCW as feed additive on rumen fermentation characteristics.

| Item | Control | 1% | 10% | 20% | SEM | *p*-Value |
|---|---|---|---|---|---|---|
| pH | 6.7 | 6.7 | 6.7 | 6.7 | 0.006 | 0.090 |
| IVDMD% | 48.8 [ab] | 50.3 [a] | 47.2 [a–c] | 42.9 [c] | 1.21 | 0.008 |
| Acetate (mM) | 164.6 | 166.8 | 166.8 | 168.2 | 3.51 | 0.080 |
| Propionate (mM) | 53.9 [b] | 55.4 [ab] | 56.0 [ab] | 56.4 [a] | 1.54 | 0.020 |
| Butyrate (mM) | 20.0 | 20.5 | 20.4 | 20.6 | 0.58 | 0.300 |
| TVFA [1] (mM) | 238.6 [b] | 242.7 [ab] | 243.1 [ab] | 245.1 [a] | 4.45 | 0.040 |
| Acetate (%) | 69.0 | 68.8 | 68.6 | 68.6 | 0.67 | 0.210 |
| Propionate (%) | 22.6 | 22.7 | 23.0 | 23.0 | 0.46 | 0.110 |
| Butyrate (%) | 8.4 | 8.5 | 8.4 | 8.4 | 0.2 | 0.820 |
| A:P [2] | 3.1 | 3.1 | 3.1 | 3.1 | 0.1 | 0.200 |

[1] TVFA: total volatile fatty acids. [2] A/P: acetate/propionate. SEM: Standard error of the mean. [a,b,c] means in the same row with different superscript differ significantly ($p < 0.05$).

**Table 5.** Effect of ensiled SCW as feed additive on rumen fermentation characteristics.

| Item | Control | 1% | 10% | 20% | SEM | *p*-Value |
|---|---|---|---|---|---|---|
| pH | 6.7 | 6.6 | 6.7 | 6.7 | 0.005 | 0.460 |
| IVDMD% | 54.7 | 56.0 | 55.3 | 53.8 | 0.84 | 0.850 |
| Acetate (mM) | 151.5 | 148.1 | 152.3 | 156.9 | 1.42 | 0.180 |
| Propionate (mM) | 66.0 | 64.7 | 63.7 | 67.7 | 1.28 | 0.750 |
| Butyrate (mM) | 25.2 [ab] | 24.7 [b] | 27.3 [a] | 27.4 [a] | 0.39 | 0.009 |
| TVFA [1] (mM) | 242.8 | 237.5 | 243.3 | 251.9 | 2.72 | 0.320 |
| Acetate (%) | 62.4 | 62.3 | 62.7 | 62.3 | 0.17 | 0.890 |
| Propionate (%) | 27.2 | 27.3 | 26.1 | 26.8 | 0.31 | 0.550 |
| Butyrate (%) | 10.4 | 10.4 | 11.7 | 10.9 | 0.16 | 0.150 |
| A:P [2] | 2.3 | 2.3 | 2.4 | 2.3 | 0.03 | 0.600 |

[1] TVFA: total volatile fatty acids. [2] A/P: acetate/propionate. SEM: Standard error of the mean. [a,b] means in the same row with different superscript differ significantly ($p < 0.05$).

*3.2. TRIAL. 2 and TRIAL. 3*

The inclusion of both the raw and ensiled SCW resulted in decreases in $CH_4$ production and total gas production compared to the control group when they were used as replacements for grass hay and concentrate (Tables 6 and 7). When hay was replaced with SCW, there were significant reductions in total gas production ($p < 0.001$, 5.4–27.2%). However, the reductions in $CH_4$ production (mL/day) were more pronounced when the concentrate was replaced by either raw or ensiled SCW, with reductions ranging from 6% to 59%, compared to the grass diet replacement groups, which showed reductions of 1.9–38% (Tables 6 and 7). The IVDMD was also significantly reduced by the inclusion of SCW, with the hay replacement groups showing reductions of 3–27% ($p < 0.001$) and the concentrate replacement groups showing reductions of 3.8–33.5% ($p < 0.001$, Tables 8 and 9). However, the lower dosage of SCW as a concentrate replacement did not show significant reductions in IVDMD in any groups (Table 9). The individual VFA production was significantly reduced in all tested groups when raw SCW was used to replace both hay and concentrate ($p < 0.001$, Tables 8 and 9), but there were no significant effects on VFA production in some of the ensiled SCW groups (Tables 8 and 9). Additionally, butyrate production was increased specifically when ensiled SCW was used to replace hay (Table 7). In contrast, when SCW was used to replace the concentrate, all groups showed significant reductions in butyrate production (Table 9). The acetate to propionate ratio (A:P) increased in all groups when raw SCW used to replace hay, except for the 70:30 raw SCW group (Table 8), while none of the ensiled SCW groups showed any significant effects. In contrast, when the concentrate was replaced with SCW, most groups showed significant increases in A:P ratio, except for the 70:30 ensiled group (Table 9).

**Table 6.** Effect of SCW as replacing grass hay on gas production profile.

| Item | Control | Raw | | | | Silage | | | | SEM | *p*-Value |
|---|---|---|---|---|---|---|---|---|---|---|---|
| | | 30:70 | 50:50 | 70:30 | 100 | 30:70 | 50:50 | 70:30 | 100 | | |
| Total gas (mL/day) | 53.9 a | 50.4 b | 47.4 c | 46.1 c | 39.2 d | 50.9 b | 51.0 bc | 49.0 bc | 44.9 d | 0.52 | <0.001 |
| Total gas/DDM [1] (mL/g) | 207.6 ab | 211.9 bc | 215.4 b–d | 208.4 b–e | 199.5 be–g | 199.4 a–c | 200.0 ab | 201.2 bcef | 188.9 g | 1.39 | <0.001 |
| $CH_4$ (%) | 4.8 | 4.6 | 4.8 | 4.7 | 4.1 | 4.7 | 5.0 | 4.6 | 4.3 | 0.09 | 0.300 |
| $CO_2$ (%) | 95.2 | 95.4 | 95.3 | 95.3 | 95.9 | 95.3 | 95.0 | 95.4 | 95.7 | 0.09 | 0.300 |
| $CH_4$/DDM (mL/g) | 10.0 | 9.8 | 10.2 | 9.8 | 8.2 | 9.4 | 10.0 | 9.2 | 8.1 | 0.19 | 0.030 |
| $CH_4$ (mL/day) | 2.6 a | 2.3 a | 2.2 a | 2.2 a | 1.6 b | 2.4 a–c | 2.6 ab | 2.2 a–c | 1.9 c | 0.05 | <0.001 |
| $CO_2$/DDM (mL/g) | 197.6 ab | 202.1 ab | 205.2 a | 198.6 ab | 191.3 b | 190.0 a–c | 189.9 a–c | 192.0 ab | 180.8 c | 1.33 | <0.001 |
| $CO_2$ (mL/day) | 51.3 a | 48.1 b | 45.2 c | 43.9 c | 37.6 d | 48.5 b | 48.5 bc | 46.8 bc | 43.0 d | 0.49 | <0.001 |

[1] DDM: Digestible dry matter. SEM: Standard error of the mean. a,b,c,d,e,f,g means in the same row with different superscript differ significantly ($p < 0.05$).

**Table 7.** Effect of SCW as replacing concentrate on gas production profile.

| Item | Control | Raw | | | | Silage | | | | SEM | *p*-Value |
|---|---|---|---|---|---|---|---|---|---|---|---|
| | | 30:70 | 50:50 | 70:30 | 100 | 30:70 | 50:50 | 70:30 | 100 | | |
| Total gas (mL/day) | 48.2 a | 45.7 bc | 42.1 d | 38.4 d | 31.0 e | 47.6 b | 45.7 bc | 41.1 d | 36.6 d | 0.65 | <0.001 |
| Total gas/DDM [1] (mL/g) | 205.0 ab | 201.7 bc | 198.3 b–d | 197.2 b–e | 183.9 be–g | 207.0 a | 204.8 ab | 194.2 bcef | 180.1 g | 1.54 | 0.004 |
| $CH_4$ (%) | 4.8 a | 4.4 a | 4.2 a | 3.9 b | 3.1 c | 4.6 ab | 3.9 b | 4.1 b | 3.6 b | 0.09 | <0.001 |
| $CO_2$ (%) | 95.2 a | 95.6 a | 95.8 a | 96.1 b | 96.9 b | 95.4 b | 96.2 ab | 95.9 ab | 96.4 a | 0.09 | <0.001 |
| $CH_4$/DDM (mL/g) | 9.7 a | 8.7 ab | 8.2 bc | 7.6 bc | 5.7 d | 9.3 ab | 8.7 a–c | 7.9 c | 6.7 d | 0.19 | <0.001 |
| $CH_4$ (mL/day) | 2.3 a | 2.0 ab | 1.8 bc | 1.5 c | 1.0 d | 2.2 ab | 2.0 b | 1.7 c | 1.3 d | 0.05 | <0.001 |
| $CO_2$/DDM (mL/g) | 191.7 | 187.8 | 187.0 | 186.3 | 181.8 | 194.4 | 192.7 | 186.2 | 178.1 | 1.42 | 0.150 |
| $CO_2$ (mL/day) | 45.9 a | 43.6 a | 40.3 b | 36.9 c | 30.1 d | 45.4 ab | 43.7 ab | 39.4 c | 35.2 d | 0.61 | <0.001 |

[1] DDM: Digestible dry matter. SEM: Standard error of the mean. a,b,c,d,e,f,g means in the same row with different superscript differ significantly ($p < 0.05$).

**Table 8.** Effect of SCW as replacing grass hay on rumen fermentation characteristics.

| Item | Control | Raw | | | | Silage | | | | SEM | *p*-Value |
|---|---|---|---|---|---|---|---|---|---|---|---|
| | | 30:70 | 50:50 | 70:30 | 100 | 30:70 | 50:50 | 70:30 | 100 | | |
| pH | 6.6 b | 6.6 ab | 6.6 ab | 6.6 ab | 6.6 a | 6.6 b | 6.6 b | 6.6 b | 6.6 b | 0.004 | 0.001 |
| IVDMD% | 58.1 a | 52.6 b | 48.3 c | 48.1 c | 42.3 d | 56.4 ab | 56.1 a–c | 53.2 cd | 51.6 d | 0.570 | <0.001 |
| Acetate (mM) | 37.2 a | 35.7 b | 34.2 c | 33.8 c | 31.5 d | 36.5 a–c | 36.2 ab | 35.2 bc | 34.2 d | 0.680 | <0.001 |
| Propionate (mM) | 14.0 a | 13.0 b | 12.1 c | 11.6 c | 10.4 d | 13.6 a–d | 13.6 a–c | 13.4 a–d | 12.8 d | 0.240 | <0.001 |
| Butyrate (mM) | 6.0 a–e | 6.0 a–e | 5.8 de | 5.9 a–e | 5.8 e | 6.0 a–d | 6.2 ab | 6.1 a–c | 6.2 a | 0.070 | <0.001 |
| TVFA [1] (mM) | 57.1 a | 54.6 b | 52.0 c | 51.4 c | 47.7 d | 55.8 a–c | 56.0 ab | 54.8 b–d | 53.1 d | 0.860 | <0.001 |
| Acetate (%) | 64.8 d | 65.0 b–d | 65.4 a–c | 65.5 ab | 65.7 a | 64.5 ab | 64.3 ab | 64.1 b | 63.9 b | 0.300 | <0.001 |
| Propionate (%) | 24.4 a | 23.7 ab | 23.1 bc | 22.6 c | 21.9 d | 24.3 a | 24.3 a | 24.5 | 24.2 | 0.110 | <0.001 |
| Butyrate (%) | 10.9 d | 11.3 c | 11.4 b | 11.9 b | 12.4 a | 11.2 bc | 11.4 bc | 11.5 ab | 11.9 a | 0.320 | <0.001 |
| A:P [2] | 2.7 d | 2.7 cd | 2.8 bc | 2.9 b | 3.0 a | 2.7 | 2.7 | 2.6 | 2.7 | 0.020 | <0.001 |

[1] TVFA: total volatile fatty acids. [2] A/P: acetate/propionate. SEM: Standard error of the mean. a,b,c,d,e means in the same row with different superscript differ significantly ($p < 0.05$).

**Table 9.** Effect of SCW as replacing concentrate on rumen fermentation characteristics.

| Item | Control | Raw | | | | Silage | | | | SEM | *p*-Value |
|---|---|---|---|---|---|---|---|---|---|---|---|
| | | 30:70 | 50:50 | 70:30 | 100 | 30:70 | 50:50 | 70:30 | 100 | | |
| pH | 6.6 d | 6.7 c | 6.7 ab | 6.7 ab | 6.7 a | 6.7 a–d | 6.7 a–c | 6.7 ab | 6.7 a | 0.01 | <0.001 |
| IVDMD% | 53.5 a | 51.2 a | 47.2 b | 43.2 c | 35.6 d | 51.5 ab | 49.8 b | 46.2 c | 42.7 d | 0.65 | <0.001 |
| Acetate (mM) | 39.3 a | 38.0 b | 37.0 b | 35.7 c | 33.5 d | 38.7 b | 38.0 bc | 37.2 cd | 36.1 d | 0.94 | <0.001 |
| Propionate (mM) | 14.2 a | 13.2 b | 12.2 c | 11.2 d | 9.7 e | 13.7 ab | 13.1 bc | 12.5 c | 11.7 d | 0.27 | <0.001 |
| Butyrate (mM) | 3.7 a | 3.4 b | 3.1 bc | 2.7 c | 2.2 d | 3.4 b | 3.2 bc | 2.9 cd | 2.5 d | 0.18 | <0.001 |
| TVFA [1] (mM) | 57.2 a | 54.6 b | 52.2 c | 49.6 d | 45.4 e | 55.8 ab | 54.3 bc | 52.6 c | 50.3 d | 1.13 | <0.001 |
| Acetate (%) | 68.1 e | 69.2 d | 70.2 c | 71.4 b | 73.2 a | 68.9 c | 69.5 b | 70.3 b | 71.8 a | 0.41 | <0.001 |
| Propionate (%) | 25.0 a | 24.3 a | 23.5 b | 22.6 c | 21.4 d | 24.7 ab | 24.4 a–c | 23.9 cd | 23.2 d | 0.16 | <0.001 |
| Butyrate (%) | 6.8 a | 6.8 ab | 6.3 bc | 6.0 cd | 5.4 e | 6.4 ab | 6.2 bc | 5.9 cd | 5.6 d | 0.36 | <0.001 |
| A:P [2] | 2.7 e | 2.9 d | 3.0 c | 3.2 b | 3.4 a | 2.8 cd | 2.9 bc | 3.0 b | 3.1 a | 0.03 | <0.001 |

[1] TVFA: total volatile fatty acids. [2] A/P: acetate/propionate. SEM: Standard error of the mean. a,b,c,d,e means in the same row with different superscript differ significantly ($p < 0.05$).

## 4. Discussion

### 4.1. Nutritive Value

In the present study, the proximate analysis showed that SCW has high nutritive value (NDF > 650 g/kg; ADF > 350 g/kg; CP > 120 g/kg). Previously, coffee grounds have been shown to contain high protein, fat and fiber levels, meaning that it can be considered as a feed source for ruminants [21]. Several studies have used coffee pulp and husk as feed for ruminants [13,22]. Additionally, some previous studies have reported that coffee residue contains high levels of organic compounds and is an appropriate substrate for fermentation processes [11,13,14]. Although it was reported that SCW is rich in fatty acids and phenolic compounds [23,24] and their significant role in altering rumen fermentation, their concentrations were not evaluated in the current study due to a shortage of material availability after completing the proximate analysis and the in vitro trials. In the forthcoming research, evaluating the content of fatty acids and phenolic compounds is strongly recommended to be considered for better understanding their mode of action.

### 4.2. SCW as a Feed Additive at the 1%, 10%, 20% Levels of DM

In the present study, the addition of raw SCW resulted in increased gas production and improved rumen fermentation parameters. The addition of raw SCW also led to increases in total VFA production and propionate production, with no changes in acetate production. These findings could be attributed to the presence of polyphenols and fatty acids in SCW, which have been shown to exhibit anti-methanogenic effects in the rumen [25–27]. However, the decrease in IVDMD with higher dosages of raw SCW was consistent with previous studies, which have indicated that plant secondary metabolites, including phenolic compounds and fatty acids, can slow intake degradation while improving ruminant production. These products can reduce the nutritive value of SCW at increased dosages but they can also exhibit other beneficial rumen modulation effects, such as reduce protein and starch degradation and inhibited amino acid degradation, via selective actions on certain rumen microorganisms. Some in vivo studies have reported that these compounds can improve live weight, milk production and ovulation rate in ruminants [27,28].

The addition of ensiled SCW resulted in increased gas production without any significant effects on the IVDMD or rumen fermentation parameters. However, there were increases in butyrate production with higher dosages of ensiled SCW. This could be attributed to the presence of certain compounds in the silage, which can affect the diversity of ruminal bacteria and the ability of certain bacterial taxa to degrade lignocellulosic material, ultimately leading to increased butyrate concentrations [29,30]. The results from both the raw and ensiled SCW groups were almost the same, but the addition of raw SCW had stronger impact on rumen parameters. In addition, both raw and ensiled SCW can be considered as good sources of energy and protein, as previously described by Senevirathne et al. (2012) [31]. Therefore, they could potentially be used as prebiotic feed additives to enhance the health status of animals.

### 4.3. SCW as a Feed Replacement

In the present study, significant reductions in $CH_4$ production were observed in almost all groups in TRAIL. 3. Previous studies have reported that coffee grounds contain significant amounts of lipids, particularly palmitic acid (C16:0) and linoleic acid (C18:2), which can contribute to reductions in $CH_4$ production [23,32]. Dietary lipids, especially medium-chain fatty acids (MCFAs) and long-chain unsaturated fatty acids (UFAs), have been shown to decrease $CH_4$ production in ruminants [27,33]. The addition of 1% fat, the most common source MCFAs, can reduce $CH_4$ production by 3.1% to 9.1% [34,35]. Similarly, McGinn et al. (2004) [36] found that the addition of sunflower oil decreased $CH_4$ production by 22% per 5% of DM. The results of the present study confirmed that SCW, with its high fat content (up to 140 g fat/kg DM), increased dietary fat concentrations by 1.8% and suppressed $CH_4$ emissions by 12.9%.

In our findings, the use of SCW to replace hay or concentrate in the diet of ruminants resulted in significant decreases in IVDMD in almost all groups. High lipid contents in ruminant diets have been shown to reduce DM, OM and fiber digestibility [33,37]. Previous studies have also reported that spent coffee waste contains a maximum of 14.7 wt% oil and high concentrations of tri- and monoglycerides (wt%), which can hinder feed particle adhesion and reduce nutrient availability for ruminal bacteria [32]. Furthermore, the decreases in total gas and VFA production could be attributed to the interference of glycerol in SCW, which lead to slow feed particle adhesion and reduced degradability [33,38].

Contrary to expectations, the addition of SCW resulted in reduced VFA production and increased pH compared to control, particularly in the concentrate replacement groups. This finding contradicted the results of some previous studies that showed little or no impact of spent coffee waste on ruminal pH or VFA production [14,31]. The reductions in VFA production and increases in pH could be attributed to certain polyphenolic compounds, such as tannins, lignans and caffeic acids, that are present in SCW. These compounds have been shown to reduce total VFA concentrations and alter ruminal microbial diversity [13,25]. These reductions in total VFA could also be related to the structures of tannin carbohydrate and protein compounds that cannot be degraded by rumen microbes or are toxic to ruminal microbes [25]. The changes in the rumen microbial community caused by the SCW could also have led to altered rumen fermentation.

In previous studies, researchers have explored the use of coffee grounds to silage as a strategy to increase the nutritional value of animal diets and address environmental concerns [11,13,14]. The ensiling process involves fermenting and preserving the coffee grounds, which potentially enhancing their digestibility, nutrient profile and storage time [11]. However, in the present study, no significant effects of ensiled SCW were observed on rumen fermentation parameters. This suggested that ensiled SCW did not have any notable impacts on rumen microbial activity, gas production or volatile fatty acid production under the specific experimental conditions of this study.

Additionally, the use of SCW (raw or ensiled) as a replacement for hay or concentrate led to significant decreases in IVDMD and total VFA production, mostly at higher dosages. However, at lower dosages of ensiled SCW (specifically 70:30 and 50:50), there were no significant reductions in some ruminal fermentation parameters. Thus, these dosages could be feasible as a replacement for traditional animal feed.

## 5. Conclusions

The study suggests that SCW has potential as an alternative ruminant feed due to its nutrient composition, reducing $CH_4$ emissions and benefiting the environment. However, high SCW doses may affect animal production, while lower doses could be viable. As a feed additive, SCW improves ruminal parameters, making it a promising prebiotic. When used as a feed replacement (70:30 and 50:50 ensilage groups), no significant effects on rumen fermentation and digestibility were observed. Thus, it is advisable not to exceed these dosages when replacing conventional feed. Further research is needed to assess SCW's impact and optimal usage methods, offering potential for sustainable feeding strategies, optimizing animal health and enhancing environmental sustainability in agriculture.

**Author Contributions:** B.B.: conceptualization, visualization, project administration, methodology, formal analysis, investigation and writing—original draft E.A.: formal analysis, conceptualization, investigation, review and editing. T.N.: validation, project administration, resources, supervision, review and editing. N.F. and M.H.: resources, review and editing. All authors have read and agreed to the published version of the manuscript.

**Funding:** This research used samples was funded by Sanyu Group Co., Ltd., Kanagawa, Japan, grant number (04292022).

**Institutional Review Board Statement:** The experimental procedures used in this study were approved by the animal care and ethics committee of the Obihiro University of Agriculture and Veterinary Medicine, Japan (approval number: 21-212).

**Informed Consent Statement:** Not applicable.

**Data Availability Statement:** The data supporting the results reported in this study are available upon request from the corresponding author.

**Conflicts of Interest:** The authors declare that they have no competing interest. The funder had no role in the design of the study, the interpretation of the data, the writing of the manuscript, or the decision to publish the results.

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
