# Peer review of "Assessment of the Impact of Coffee Waste as an Alternative Feed Supplementation on Rumen Fermentation and Methane Emissions in an In Vitro Study"

_fermentation, doi:10.3390/fermentation9090858_

Round 1
Reviewer 1 Report
* In my opinion authors should change the word effect on the title try to use other word hence it was so common
*Line 20 authors mentioned ( two different experimental) while at line 25 (TRIAL. 3), Please clarify
*Line 22 spent coffee waste should be SCW , the same for all manuscript , Please
*Keywords should be rearranged according to alphabetical order
*Line 22 fresh matter of what? clarify please
*Line 62 methane should be CH4
*Line 83 in vitro should be an italic
*Introduction lack of clear hypothesis
*Line 101 sprayed with vinegar spray, ? why?
*Line 113 collected rumen fluid, How did you collect? clarify in Details please
*Line 114 Thermos flask . flushed with CO2 or not?
*Needs animals welfare and ethical protocol number for animal use
* Lines ( 121 - 140) Please the description of the experimental design and treatment appear very confused for the readers so try to rewrite with more simplesity
*Line 140 what about the bottles for internal stander?
*Line 141 artificial saliva or nutritive solution ?
* what about the bottles capacity ?
* line 146 using a gas-tight syringe, clarify ? How many you collect the gas?
* headspace how many ml?
*Line 194 there was a tendency. In my opinion i did not prefer to use tendency once you did not mentioned in your statistical section ,
*The authors use significant word without showing the p value why?
*Table 1 needs to put ash
* The results section needs some improvement for shorten it please once all values excite on tables
*Table 2 I would prefer if the authors calculated CH4 according to DOM not CH4 /DDM (ml/g) it will be more accurate without including the ash in the calculation
*Table 6,7,8, and9 need more improvements , so much values if authors could present the tables with other form it will be better
* In my point of view i would prefer that the authors should analyze the main active components in SCW such *( phenolic compounds and fatty acids) because it will be so useful for the discussion part
*Line 252 in vivo should be an italic
* The discussion part , i see that authors depended on his discussion on the main active ingredient in SCW but such data is not showing in this study , so its a missing important point
* The conclusion is so lone need to shorten please
*References , authors needs to exclude all old references please try to replace be recent one
Quality of English Language looks fine with minor revision
Author Response
Response to Reviewer– 1
* In my opinion authors should change the word effect on the title try to use other word hence it was so common
Assessment of the impact of coffee waste as an alternative feed supplementation on rumen fermentation and methane emissions in an in vitro study
*Line 20 authors mentioned ( two different experimental) while at line 25 (TRIAL. 3), Please clarify
We have used two experimental designs, in this case TRIAL. 2 and 3 are the same experimental design.
*Line 22 spent coffee waste should be SCW , the same for all manuscript , Please
We have converted the spent coffee waste to SCW; please locate it in L 22.
*Keywords should be rearranged according to alphabetical order
We have ordered the keywords alphabetically; please find them in L 34.
*Line 22 fresh matter of what? clarify please
Fresh matter of basal diet: please find the change in L 22 of the manuscript, which we made.
*Line 62 methane should be CH4
We have converted the methane to CH4; please locate it in L 62.
*Line 83 in vitro should be an italic
We've shifted to italics, so please find it in L 90.
*Introduction lack of clear hypothesis
The introduction section of the manuscript was improved, and the hypotheses were added. Find it in L 95-99, please.
*Line 101 sprayed with vinegar spray, ? why?
Sprayed with vinegar is the treatment method used in Starbucks Japan stores, according to the manual. It is intended to have an antiseptic effect. It has not been completely proven scientifically, but it is the procedure in the field.
*Line 113 collected rumen fluid, How did you collect? clarify in Details please
Please refer to L 131-134 for additional information regarding the collection of rumen fluid that we have added.
*Line 114 Thermos flask . flushed with CO2 or not?
While collecting rumen fluid, CO2 was not flushed into the flask; nevertheless, once the fluid was in the flask, we used vacuum gap and the flask was full.
*Needs animals welfare and ethical protocol number for animal use
Please find in the methodology, the protocol that we were referring to in L 120-124.
* Lines ( 121 - 140) Please the description of the experimental design and treatment appear very confused for the readers so try to rewrite with more simplesity
Please see the experimental design from L 136-156 in the revised version of the manuscript that we attempted to create.
*Line 140 what about the bottles for internal stander?
Some bottles were used for control groups, some bottles were used as blanks for each run, and some bottles were utilized for experimental treatments. Please find it in L 153-156.
*Line 141 artificial saliva or nutritive solution ?
This was artificial saliva used L 161
* what about the bottles capacity ?
This information is provided in the L 161. The fermentation bottle had a 120 mL capacity and made by glass.
* line 146 using a gas-tight syringe, clarify ? How many you collect the gas?
* headspace how many ml?
Total gas from the 24h incubation was collected using a gas-tight syringe, and subsamples of 3 ml from each sample were preserved for further analysis in a vacutainer tube. Please find it in L 166-168
*Line 194 there was a tendency. In my opinion i did not prefer to use tendency once you did not mentioned in your statistical section ,
In the statistics section of the text, we inserted a note concerning tendency. Please find it in L 200-201
*The authors use significant word without showing the p value why?
p values for L 207, 210, 229, 234, 235, 239, 290, 301, 329, and 331 were included in the paper.
*Table 1 needs to put ash
We put new line for crude ash in the table 1
* The results section needs some improvement for shorten it please once all values excite on tables
We agreed with the comment, and the authors made an effort to condense it in the result section. Please find it from L 205-215 and 226-242.
*Table 2 I would prefer if the authors calculated CH4 according to DOM not CH4 /DDM (ml/g) it will be more accurate without including the ash in the calculation
We agree with your suggestion that CH4 should be calculated according to DOM, but we are only doing a preliminary analysis at this time. So, we did not consider the digestibility of the organic matter in this trial.
*Table 6,7,8, and9 need more improvements , so much values if authors could present the tables with other form it will be better
Authors would want to make tables seem better, but we wouldn't want to change the information in the tables because it would be useful to readers. Table 2, 3, 4, 5, 6, 7, 8, 9
* In my point of view i would prefer that the authors should analyze the main active components in SCW such *( phenolic compounds and fatty acids) because it will be so useful for the discussion part
* The discussion part , i see that authors depended on his discussion on the main active ingredient in SCW but such data is not showing in this study , so its a missing important point
We totally agree with your perspective on the importance of evaluating the contents of phenolic compounds and fatty acids. However, due to the shortage of material available after completing the proximate analysis and these several in vitro trials, we couldn’t perform those analyses. We know this is a shortcoming in this study, and we believe this should be considered in our future trials; therefore, we highlighted this matter in L 256-261.
*Line 252 in vivo should be an italic
This word has been changed to an italic please find it L 274
* The conclusion is so lone need to shorten please
We agree with this point, we have shorten that section. Please find it in L 334-343
*References , authors needs to exclude all old references please try to replace be recent one
We tried to update the references. Please find it in 403-405, 415-417, 422-436, 466-468
Reviewer 2 Report
Specific comments and questions:
The presented article aims to assess the level of incorporation and method of utilization of Spent Coffee Waste (SCW) as ruminant feed and to study its effects on in vitro rumen fermentation characteristics and methane (CH4) production. At the outset, it is necessary to add that SCW is a technological waste product that is a worldwide renewable source, cheap and sustainable. Some authors try to use this product in a strategy to turn waste products into usable resources, which is a necessity for the sustainable future of our planet. Furthermore, the research characterization of this waste product as an alternative feed source is important from a nutritional, ecological and circular economy point of view. In other words, the article deserves attention.
The following specific comments and questions are considered as important:
In the introduction presentation:
· Information on ethical statements is lacking.
· Given the importance of fatty acids and polyphenols as a constituent of SCW, it would be good to include them in the chemical composition analysis.
Several suggestions are made in the description of Materials and Methods:
· My consideration is that the text included in subsection 2.4 from l.117 to l.120 and from l.141 to l.144 have to be moved to the beginning of the next subsection 2.5. In this case, the subtitle of 2.4 should become just “Experimental design”.
· The proper place of Table 1 is immediately after l.95.
In the Results presentation:
· There is some inconsistency between the description of the results and the numbering of Tables 2, 3, 4 and 5. It would be good if all these tables were moved immediately after l.200., before the description of the results of TRIAL 2 and TRIAL 3.
· It would be good for the authors to think about the order of presentation of the results obtained in TRIAL 3 (Tables 6,7, 8 and 9) - first the replacement of parts of the hay (according to the experimental design) or the concentrate mixture (as it is now)?
· l.202 - add clarification that this affects SCW when replacing part of grass hay (TRIAL 2) and concentrate mixture (TRIAL 3).
· l. 205 - for CO2, please double check the results and the text, now there is a discrepancy between them.
· l. 213- insert ``individual'' before VFA cont.
· l.217-222 the text describing the results should double check against the digital information in the table, make the description more precise.
· The subsections of section 3. (Results) are better numbered as 3.1 and 3.2 for TRIAL 1 and TRIAL 2 and TRIAL 3, respectively.
· in the title of Table 1, the unit (g/kg dry matter) should be deleted as it is present on the next line
· The title of Tables 2, 3, 4 and 5 needs from a little correction- “Effect of ... SCW as A feed additive on...”
The Discussion (and all its subsections) and Conclusion sections are numbered incorrectly, please correct them.
A discussion is given which is very detailed and its main parts are consistent with the experimental design: nutritional value of the SCW as an experimental feed, different involvements of raw or ensiled SCW in the substrate dry matter and different replacements of grass hay or concentrate with raw or ensiled SCW. There is also a general discussion, which is positive point.
From the results of the study, the authors draw conclusions that follow the research objectives.
· To be relevant and acceptable, they need to be edited precisely and reformulated – to be specific, not descriptive, short and to the point.
Author Response
Response to Reviewer– 2
The following specific comments and questions are considered as important:
In the introduction presentation:
- Information on ethical statements is lacking.
Information added as in L 120-124.
- Given the importance of fatty acids and polyphenols as a constituent of SCW, it would be good to include them in the chemical composition analysis.
We added more information about the fatty acids and polyphenols of SCW in the introduction part. Please find it in L 66-76.
For the chemical composition, we totally agree with your perspective on the importance of evaluating the contents of phenolic compounds and fatty acids. However, due to the shortage of material available after completing the proximate analysis and these several in vitro trials, we couldn’t perform those analyses. We know this is a shortcoming in this study, and we believe this should be considered in our future trials; therefore, we highlighted this matter in L 256-261.
Several suggestions are made in the description of Materials and Methods:
- My consideration is that the text included in subsection 2.4 from l.117 to l.120 and from l.141 to l.144 have to be moved to the beginning of the next subsection 2.5. In this case, the subtitle of 2.4 should become just “Experimental design”.
We followed your suggestions L 135, 136-156, 158-164
- The proper place of Table 1 is immediately after l.95.
We followed your suggestions L 107
In the Results presentation:
- There is some inconsistency between the description of the results and the numbering of Tables 2, 3, 4 and 5. It would be good if all these tables were moved immediately after l.200., before the description of the results of TRIAL 2 and TRIAL 3.
All tables were relocated to after the description of TRIAL. 1 and renumbered L 204-216.
- It would be good for the authors to think about the order of presentation of the results obtained in TRIAL 3 (Tables 6,7, 8 and 9) - first the replacement of parts of the hay (according to the experimental design) or the concentrate mixture (as it is now)?
The tables was changed; Table 6-9, L 220- 223
- l.202 - add clarification that this affects SCW when replacing part of grass hay (TRIAL 2) and concentrate mixture (TRIAL 3).
Please see L 226-241 for the updated result part of TRIAL 2 and 3.
- l. 205 - for CO2, please double check the results and the text, now there is a discrepancy between them.
We have checked texts
- l. 213- insert ``individual'' before VFA cont.
We insert word that individual into L 237
- l.217-222 the text describing the results should double check against the digital information in the table, make the description more precise.
Please find the rewritten paragraph for this part in L 240-242.
- The subsections of section 3. (Results) are better numbered as 3.1 and 3.2 for TRIAL 1 and TRIAL 2 and TRIAL 3, respectively.
We renumbered it; please find it between L 204, 225.
- in the title of Table 1, the unit (g/kg dry matter) should be deleted as it is present on the next line
(g/kg dry matter) was deleted from the title of Table.1
- The title of Tables 2, 3, 4 and 5 needs from a little correction- “Effect of ... SCW as A feed additive on...”
We tried to improve Tables title for 2, 3, 4 and 5
The Discussion (and all its subsections) and Conclusion sections are numbered incorrectly, please correct them.
We were renumbered subsections please find it from L 248, 249, 262, 288, 333
Round 2
Reviewer 1 Report
Thanks for the authors no more comments , Just the In Vitro on the title should be an italic format
Author Response
* Thanks for the authors no more comments , Just the In Vitro on the title should be an italic format
The title has been changed, please find it in L4
Reviewer 2 Report
Specific comments and questions:
The presented article aims to assess the level of incorporation and method of utilization of Spent Coffee Waste (SCW) as ruminant feed and to study its effects on in vitro rumen fermentation characteristics and methane (CH4) production. At the outset, it is necessary to add that SCW is a technological waste product that is a worldwide renewable source, cheap and sustainable. Some authors try to use this product in a strategy to turn waste products into usable resources, which is a necessity for the sustainable future of our planet. Furthermore, the research characterization of this waste product as an alternative feed source is important from a nutritional, ecological and circular economy point of view. In other words, the article deserves attention.
The following specific comments and questions are considered as important:
In the Results presentation:
· It would be good if Tables 6, 7, 8 and 9, in which the results of TRIAL 2 and TRIAL3 are presented, were moved immediately after l. 247 (after the text of 3.2).
· l.241-247 please put the numbers of Tables in the brackets.
A discussion is given which is very detailed and its main parts are consistent with the experimental design: nutritional value of the SCW as an experimental feed, different involvements of raw or ensiled SCW in the substrate dry matter and different replacements of grass hay or concentrate with raw or ensiled SCW. There is also a general discussion, which is very good.
From the results of the study, the authors draw conclusions that follow the research objectives.
Author Response
The following specific comments and questions are considered as important:
It would be good if Tables 6, 7, 8 and 9, in which the results of TRIAL 2 and TRIAL3 are presented, were moved immediately after l. 247 (after the text of 3.2).
Table 6, 7, 8 and 9 moved to after the text of result TRIAL 2 and 3
- l.241-247 please put the numbers of Tables in the brackets.
Renumbered the tables in the brackets